# Orchestrating Treatment Modalities in Metastatic Pancreatic Neuroendocrine Tumors—Need for a Conductor

**DOI:** 10.3390/cancers14061478

**Published:** 2022-03-14

**Authors:** Alexander R. Siebenhüner, Melanie Langheinrich, Juliane Friemel, Niklaus Schaefer, Dilmurodjon Eshmuminov, Kuno Lehmann

**Affiliations:** 1Clinic for Gastroenterology and Hepatology, University Hospital Zurich and University of Zurich, Rämistrasse 100, CH-8091 Zurich, Switzerland; 2ENETS Center of Excellence Zurich, Rämistrasse 100, CH-8091 Zurich, Switzerland; kuno.lehmann@usz.ch; 3Department of Visceral Surgery, University Hospital Greifswald, Ferdinand-Sauerbruch-Strasse, D-17475 Greifswald, Germany; melanie.langheinrich@med.uni-greifswald.de; 4Institute for Pathologie, University Bern, Murtenstrasse 31, CH-3008 Bern, Switzerland; juliane.friemel@pathology.unibe.ch; 5Department of Nuclear Medicine, University Hospital Lausanne, Rue du Bugnon 46, CH-1011 Lausanne, Switzerland; niklaus.schaefer@chuv.ch; 6Department of Surgery and Transplantation, University Hospital of Zurich, Rämistrasse 100, CH-8091 Zurich, Switzerland; dilmurodjon.eshmuminov@usz.ch

**Keywords:** neuroendocrine tumors, pancreatic, liver metastasis, surgery, liver transplantation, timing

## Abstract

**Simple Summary:**

Pancreatic neuroendocrine tumors (pNET) are a heterogeneous and challenging entity, and today’s guidelines offer a variety of treatment modalities, while surgery has a clear role for patients with resectable tumors and early stages, advanced, or metastatic pNET may benefit from treatments that were evaluated in randomized controlled studies during the last year. With this review, we aim to provide an updated view on treatment options for metastatic pNET.

**Abstract:**

Pancreatic neuroendocrine tumors (pNETs) are a vast growing disease. Over 50% of these tumors are recognized at advanced stages with lymph node, liver, or distant metastasis. An ongoing controversy is the role of surgery in the metastatic setting as dedicated systemic treatments have emerged recently and shown benefits in randomized trials. Today, liver surgery is an option for advanced pNETs if the tumor has a favorable prognosis, reflected by a low to moderate proliferation index (G1 and G2). Surgery in this well-selected population may prolong progression-free and overall survival. Optimal selection of a treatment plan for an individual patient should be considered in a multidisciplinary tumor board. However, while current guidelines offer a variety of modalities, there is so far only a limited focus on the right timing. Available data is based on small case series or retrospective analyses. The focus of this review is to highlight the right time-point for surgery in the setting of the multimodal treatment of an advanced pancreatic neuroendocrine tumor.

## 1. Introduction

Neuroendocrine tumors (NET) are a heterogeneous group of tumors with primary origin often located in the gastro-entero-pancreatic (GEP) tract. Although the incidence has increased significantly in recent years, these tumors are still considered a rare entity. [1]. Prognosis and biological behavior are mainly driven by the primary tumor site, the growth index of the tumor cells (Ki-67, mitotic count), and the primary tumor stage at diagnosis [2,3,4,5,6]. Herein, pancreatic neuroendocrine tumors (pNET) show the worst prognosis at any stage or grade compared to midgut NETs [1,7], particularly if p-NETs represent liver metastasis [8]. Of note, more than 60% of pNETs present in the advanced or metastatic stage at first diagnosis [9].

Over 40% of pNET become metastatic in the course of the disease, commonly by lymph nodes or liver metastasis [10], and patients with untreated liver metastasis show a 5-year survival rate of between 20% to 40% [11,12]. Compared to gastrointestinal NET, pNET rarely presents with classical neuroendocrine symptoms [13]. Usually, the initial presentation of pNET is unspecific unless the tumor is hormonally active or the patient represents extensive symptomatic liver metastasis.

Different treatment modalities exist for advanced pNETs reflected by the treatment recommendation of several current guidelines (e.g., ENETS, NCCN, ESMO) [14,15,16]. Discussion in a multidisciplinary tumor board (MTD), or even a dedicated NET center, is highly recommended. Patients should be presented in MTDs repetitively over time, since moderate to high differentiated pNETs likely receive multiple treatment modalities and sequences due to their course of disease [3,17]. So far, evidence about the timing of these steps is limited and based on retrospective and center-based experience [18,19,20,21]. With the advent of recent systemic treatment options, including PRRT, the prognosis of metastatic NET has improved over the last few years. This has opened a window for additional locoregional treatment in selected patients who remain stable over time. The key issue for sequential treatment steps, however, remains the right timing. With this review, we address the question of the timing of modalities in the setting of advanced or metastatic pNETs treatment in the European Neuroendocrine Tumor Society Center of Excellence (ENETS CoE).

## 2. Methods

This review was written as a narrative review. The objective was to highlight the right time-point for surgery in the setting of advanced pancreatic neuroendocrine tumors, as well as illustrating key biomarkers that could be applied for a specific scenario. Literature research was made via Pubmed in July 2021 using the terms “advanced pancreatic neuroendocrine tumor”, “liver metastasis”, “surgery”, “downstaging”, “targeted therapies/multikinase inhibition (MKI)”, “peptide receptor radionuclide therapy (PRRT)”, “immune checkpoint inhibitors (ICI)”, “immunotherapy”, and “systemic therapy”. Additional references were retrieved from articles. As this was not a systematic review, no formal inclusion/exclusion criteria were selected. However, we cited studies that provided information regarding the evaluation of surgery under systemic treatment as PRRT or MKIs or ICIs, focusing on downstaging and enabling surgery in this sequence in pNET. We also selected studies referring to the follow-up of surgery in advanced pNET. We did not focus on the side effects or toxicities of these treatments as the time-point of surgery in advanced pNET was used for key evaluation. When possible, we highlighted the survival rates as overall survival (OS) and/or progression-free survival (PFS) in the groups when surgery was used.

## 3. Results

### 3.1. Prognostic Factors in Metastatic Pancreatic NET

Chromogranin A (CgA), Synaptophysin and neuron-specific enolase (NSE) are widely used in clinical routines as diagnostic tumor markers. Since both CgA and NSE have a wide range in sensitivity (CgA 43–100%; NSE between 33% and 59%), as well in specificity (CgA 10–96%; NSE up to 80%) [22,23], further markers are currently investigated in clinical routines [24,25,26]. Regarding any multimodal treatment, it is important to emphasize that the histopathological workup of pNETs should include essential prognostic (tumor stage, grade, nodal involvement) and predictive factors (SSTR2 status) to guide a risk stratification. These risk groups differ in terms of tumor biology and clinical behavior and are likely to be responders or non-responders to specific treatment modalities, e.g., targeting the somatostatin receptors if expressed. SSTR2 immunohistochemistry correlates with SSTR2 imaging; tumors with expression in >10% of tumors cells were shown to be suitable for in vivo targeting [27]. The revised cut-off for grading pNETs according to WHO 2017 or ENETS classification, uses <3% proliferation index as the cut-off for G1, 3–20% for G2, and >20% for G3 tumors (Table 1) [28,29]. Defining criteria for neuroendocrine carcinoma (pNEC) are small (oat)-cell or large-cell morphology, with proliferation rates of usually >50%. DAXX/ATRX immunohistochemistry staining in tumor cells is used as a tissue-based biomarker in non-metastasized settings, especially G2 tumors with potential progressive behavior. A loss of DAXX/ATRX is associated with chromosome instability and reduced survival [27]. Recently, a meta-analysis of 14 studies with a total of 2313 patients has supported the prognostic significance of altered DAXX/ATRX genes in pNETs with a combined HR of 5.05 for disease-free survival (95% CI: 1.58–16.20, *p* = 0.01) [30,31]. In metastatic disease, DAXX/ATRX loss seems to be associated with longer survival. Due to the tumor heterogeneity of pNET per se, the Ki-67 index of the metastatic lesion may differ from that of the primary lesion. If this higher lesion won’t be detected due to the histologic workup by only taking one or two biopsies, the patient may be undertreated with a negative effect on their survival [32]. Therefore, multiple biopsies from primary, as well from metastasis, should be considered [33,34] for precise detection of the proliferation status. Table 2 summarizes the prognostic factors for surgery.

### 3.2. The Orchestra of Treatment Modalities for Metastatic Pancreatic NET

#### 3.2.1. Surgery and Locoregional Treatment

Surgery remains the fundamental locoregional modality for resectable pNET and offers a chance for cure. The most frequent metastatic site of pNET is the liver due to portal venous drainage of the tumor. For technically resectable metastatic pNET with a favourable G1/G2 differentiation, resection of all manifestations remains the primary modality, recommended also by the neuroendocrine tumor societies [35,36]. In the setting of non-resectable metastatic pNET, however, resection of the primary is controversially discussed. The major argument, particularly for duodeno-pancreatectomy and, to a lesser extent, for left resection, is the unclear impact on survival and the potential morbidity and mortality of the procedure in the setting of a metastatic pNET.

In the setting of metastatic pNET, two scenarios should therefore be separated: liver-only and extrahepatic metastasis. For liver-only metastasis, a variety of strategies are available to achieve resectability in borderline situations. Traditionally, two-stage procedures, including portal vein embolization or ligation, enable safer major liver metastasis resections in patients with too-small remnants after hypertrophy [37]. These procedures allow the remnant to grow after selective occlusion of the contralateral portal vein and can be combined with minor liver resections to clear the future remnant. In a second step, major liver surgery is performed to achieve a radical resection. Recently, this concept has evolved and portal vein ligation has been combined with staged tissue transection, which further enhanced the regenerative capacity of the remnant liver and pushed the border of resectability in NET patients [38]. Although accomplished by specialized teams, these advanced resection techniques ultimately fail to overcome a serious limitation of liver surgery: the incidence of hepatic relapse of pNET. Recurrence from hepatic metastasis tends to occur within the first year and occurs in up to 94% of pNET patients, despite an excellent OS (74% at 5 and 51% at 10 y) [39]. This high rate of recurrence is mostly due to microscopic disease, which tends to be largely underestimated by the current preoperative standard, contrast-enhanced MR, as shown in an elegant study comparing imaging with thin-slice histopathology [40]. To overcome this clinical challenge, liver transplantation might be evaluated for selected patients and tumor characteristics (e.g., low-grade NET) with excellent survival rates [41]. The potential benefit of liver transplantation for metastatic NET depends on a stringent patient selection. Thus, many patients disqualify due to disease progression under systemic treatment. Many patients also have associated contraindications to liver transplantation (e.g., age, portal hypertension, or co-presentation of metastasis in other organs such as bone, peritoneal, or lung metastasis). Recommended selection criteria for liver transplantation includes liver metastasis from well-differentiated NET with portal vein drainage, resected primary tumor, hepatic disease load < 50%, age < 55 years, and stable disease for more than 6 months [42].

Considering it as a locoregional treatment, SIRT (selective internal radiation therapy) may contribute to controlling diffuse liver metastasis in a patient with a non-resectable disease or who does not qualify for surgery. A multi-institutional analysis of 244 NET patients demonstrated about 20% objective response rates and observed stabilization of the disease in a majority of patients [43]. However, this study lacks an adequate control group.

In the setting of extrahepatic metastasis from pNET, the disease may be present in bones, the peritoneum, or any possible anatomic site. Oligosymptomatic disease may be treated with surgery or any alternative ablative technique [44]. In the setting of peritoneal metastasis, radical surgery may be considered in highly selected patients, reflected by a multi-institutional series of 127 patients in 53 centers [45] with reasonable results.

#### 3.2.2. Role of Interventional Radiology with Locoregional Liver Therapies

Locoregional liver therapies play an important role in the management of patients with NELM, especially as they have a predominant arterial vascularization. Transarterial embolization (TAE), transarterial chemoembolization (TACE), and selective internal radiation therapy (SIRT) are intra-arterial therapies available for these patients in order to improve symptoms and overall survival. These treatment options are proposed in patients with NELM not responding to systemic therapies and without extrahepatic progression or a contraindication for surgery. Although the modalities will be regularly discussed among a multi-disciplinary team (MDT) during a tumor board, the referred data of all modalities (TAE, TACE, SIRT) rely on retrospective analysis in highly selected cases [46]. In fact, all three modalities focus on the local control of NELM; a direct comparison among the three different options is hardly possible due to the heterogeneity to include a pNET to TAE or TACE or even SIRT [47]. A therapeutic response will be estimated by summarizing the largest retrospective studies in this field [48,49,50,51,52,53]. With TAE or TACE, symptom improvement is achieved in 60% to 90% of patients and mass effect to liver involvement decreases in 100% of patients with NELM [48,49,50,51,52,53]. Interestingly, there is no direct comparison between TAE or TACE and these data rely on retrospective data only. There is some evidence that NELM from gastric or enteric NET show a better response to TAE, whereas pNET might have a higher response to TACE [54]. Of note, TAE and TACE show very low post-treatment mortality ranging from 0–8%, with most deaths related to toxic carcinoid syndrome or liver failure and the highest mortality rate related to emergency procedures of highly symptomatic NELM patients [52,55,56]. Regarding OS, the heterogeneity in the design of published studies does not allow any firm conclusion. OS ranged from 12 to 84 months after TAE or TACE and TACE achieved the highest OS in PNET [54,57,58,59] With regard to SIRT, no multicentric prospective controlled trial is available. Two retrospective meta-analyses, including more than 800 patients out of 19 studies, are available and report a median OS of more than 28 months (range: 14–70 months) after SIRT [60,61].

#### 3.2.3. Systemic Therapy

Within the last decade, various systemic treatments for pNETs emerged, offering better disease control. In well-differentiated metastatic pNET, (G1 and G2), which usually correlates with a higher expression of somatostatin receptors, somatostatin analogues (SSA) should be considered in the first line [14]. Other targeted drugs approved in the setting of well to moderated differentiated pNET are mechanistic targets of rapamycin (mTOR) inhibitors, tyrosine kinase inhibitors (TKIs), or multikinase inhibitors (MKIs), as well as some cytotoxic regimens such as temozolomide and capecitabine [62,63,64]. Most of these drugs were studied in placebo-controlled trials and resulted in better disease control by the active compound. However, data on sequential systemic treatment for pNETs is still limited and presented heterogeneous in current guidelines. In addition, predictive biomarkers for therapy guidance are an unmet need. It is important to realize that most systemic options in G1–G2 pNETs only stabilize tumor burden and improve progression-free survival (PFS). The impact on overall survival (OS) is, however, limited. Most systemic therapies do not induce a significant tumor response with complete or nearly complete remission—as known from the systemic treatment of colorectal metastasis—which would open the door for downsizing strategies and surgical interventions. An overview of systemic therapy options is shown in Table 3.

Some patients may not qualify for or may not be willing to undergo any kind of surgery. In this situation, control of liver metastasis by PRRT or SIRT should be considered an option in addition to or within a sequential approach to the installed systemic treatment [75]. In contrast to SIRT, the SSTR-2 density of the tumor cells is essential for performing a PRRT in advanced pNETs.

### 3.3. The Right Timing in Pancreatic Well Differentiated NET with Liver Metastases (NELM)—Adagio Con Moto (Slowly into Movement)

The few critical factors, which influence the choice of the first modality in the setting of metastatic pNET are tumor biology, reflected by the grading, and SSTR density— SSTR2 density mostly drives the susceptibility of pNET to somatostatin receptor therapies. A large box of options applies to the group of well-differentiated pNETs, which include G1/G2, with a Ki-67 index of up to 20%. In this situation, resectability of the primary together with metastatic lesions is critical. Currently, there is no standardized staging system for metastatic pNET that would enable the separation of a less advanced metastatic stage from a more advanced one. For practical reasons, this review will differentiate (a) resectable oligometastases from (b) extensive but liver-only metastases or (c) non-resectable extrahepatic metastases with or without liver metastases.

It is beyond the scope of this review to resume in detail all the staging modalities of metastatic pNET. With the mindset of the surgeon, which is clearly on resectability, we highlighted the role of somatostatin receptor imaging to exclude extrahepatic metastasis and to assess the potential for PRRT. In addition, contrast-enhanced MRI is highly recommended to assess the distribution, relation to hepatic vessels, and, finally, the respectability of hepatic metastasis. Particularly for large pNET, a diagnostic laparoscopy may be considered, since small peritoneal metastasis is not always visible on imaging.

For neuroendocrine liver metastasis (NELM), Frilling et al. described three different types of patterns [76]: type I shows an isolated single lesion of any type, type II has a large focus of metastatic bulk with smaller surrounding lesions involving both hemi livers, and type III describes a widely disseminated metastatic situation with the involvement of both Hemi livers and essentially no normal liver parenchyma appreciable on preoperative imaging.

Only patients with type I and selected patients with type II metastasis are candidates for upfront hepatic resection. In general, about 15% to 50% of patients with NELM might be eligible for some type of surgical procedure [38,76].

#### 3.3.1. pNET with Low Volume Liver Metastasis

Resection of the hepatic disease remains the solid fundament in the treatment of patients with resectable oligometastasis, and typically includes type I liver metastasis according to Frilling [76]. Oligometastatic liver metastasis from pNET, depending on their location and size, can usually be resected upfront, together with the primary tumor [77]. Usually, such a strategy is considered curative. Due to the lack of randomized controlled trials, the role of adjuvant treatment remains controversial after radical resection. So far, all available guidelines consider no adjuvant therapy due to the lack of evidence [14,35,36]. In the setting of oligometastatic disease, extrahepatic lesions are not a limiting factor. Resection of extrahepatic metastasis in well to moderate differentiated pNET is associated with acceptable outcomes in selected cases [17]. In the pNET setting of a clearly resectable primary tumor and its metastasis, no available data justifies a non-surgical treatment option upfront [78]. On the other hand, surgery of the primary tumor in the setting of irresectable NELM prolongs survival—but data are still interpreted with caution due to small sample sizes and selection bias [79].

#### 3.3.2. pNET with Extensive (<50%), but Confined Liver Metastasis

More than 50 percent of pNETs with NELM present with bilateral disease. This group includes patients with an extensive hepatic disease load, still below 50% of the liver volume. The extrahepatic disease is usually excluded by somatostatin receptor imaging, e.g., a ^68^Gallium DOTATATE-PET CT. Since the boundaries of technical resectability are constantly pushed forward—we mentioned different two-step procedures for liver surgery above—this group is not exclusive, compared to the other two scenarios. It is, however, important to retain the problem of multiple, non-detectable metastasis in the liver [40] when the strategy is planned. Consequently, the key question is the impact on a patient’s long-term perspective.

A given patient may follow the surgical road, which would be resection, probably several times, at later stages in combination with other ablation techniques, radiofrequency, microwave ablation, or SIRT. The additional use of SSA or any other systemic treatment short after resection is still uncertain due to the lack of comparison trials to address this question. The slow course of well-differentiated pNET offers a multiplicity of treatment options. Most important are prognostic factors as described in Table 2, whereas head-to-head comparison or the evidence of the right timing are still missing. A surgical approach is therefore preferentially indicated in situations when a biologically benign behavior is expected. As an alternative, the same patient may receive systemic treatment for disease control as a first step, which will be followed by resection of the primary in case of systemic control for several months, whereas we will consider a time interval for stabilization for at least for 6 months or longer in our center.

Primary resection may then follow another course of systemic treatment to confirm the low-grade biology of the disease. Finally, such a patient with a biologically benign disease limited to the liver may proceed to liver transplantation with the goal of total and prolonged disease control. Of course, these pathways are not mutually exclusive, and crossing is theoretically possible. It can be, however, technically be very demanding to perform safe liver transplantation after very extensive liver resections. The key question remains which pathway—resection or transplant—will lead to better or longer disease control.

Extended surgical resection for locally advanced and metastatic pancreatic endocrine tumors is feasible with encouraging disease-specific survival of up to 5 years for a majority of these selected patients [80]. In this scenario, hepatic resection of NELM will frequently involve non-anatomic resections, with most patients undergoing multiple wedge resections to debulk multifocal, bilateral disease [39,81]. Recurrence after curative resection of liver metastasis is common but may not be as frequent as published in past decades. A recent study including 481 patients found recurrence in 46% of patients, including 71% early and 29% late recurrences. On multivariate analysis, pancreatic NET, primary tumor lymph node metastasis, and a microscopic positive surgical margin were independent risk factors for early intrahepatic recurrence. Early recurrence was associated with worse disease-specific survival than late recurrence, which was 75% at 10 years. Redo-surgery improved survival to 54% at 10 years for early and late recurrence [82]. Treatment of multifocal and bilateral resection of NELM is often combined with ablation in the situation of multifocal and bilateral tumor involvement in the liver. In fact, ablation is used in up to 20 percent of multifocal surgeries in NELM [39].

Interestingly, in contrast to the high rates of R0 resection of colorectal liver metastasis, the resection rate of NELM seems much worse [39]. Associating liver partition and portal vein ligation for staged hepatectomy (ALPPS) is a novel strategy in the treatment of NELM with multiple liver metastases. ALPPS appears to be a suitable strategy for well-selected patients with NELM. However, the high rate of disease recurrence should call for careful patient selection and discussion of alternatives [38]. Table 4 gives an overview of liver resection for advanced hepatic metastasis of pNET.

Given the high recurrence rate after resection of liver metastasis and the benign behavior of well-differentiated NET, orthotopic liver transplantation (OLT) gained attendance as a radical therapy. Due to the lack of long-term results and prospective trials, the selection criteria are still poorly defined. Selection criteria, such as the Milan-NET criteria [41] or the ENETS guidelines [15], provide some reference. Table 5 lists a summary of the advantages of OLT in pancreatic NET. There was no randomized study comparing OLT versus other treatment modalities [89,90]. Favorable criteria used for selection to OLT include age (<45–55 years old), low–moderate Ki-67 Index (<10%), primary tumors solely draining into the portal system, low hepatic tumor load (<50%), and absence of EHD [42,91,92,93]. Following these favorable selection factors for OLT, patients achieve 5-year OS up to 70–90%, as well as 5-year PFS around 80% [94]. Experts in this field recommend a follow-up under systemic treatment for at least 6–12 months and postpone OLT for a biologic favorable group selection, as these patients would have a better prognosis after OLT [41].

Milan-NET criteria [41], ENETS guidelines [15] and the Organ Procurement and Trans-plantation Network in the United States highlight this statement as a recommendation [98]. Still for debate and without a clear statement is the situation whether OLT should be offered only to patients with stable disease, or even to patients with progressive disease for a rescue option, especially if the tumor grading is favorably low. The selection process of patients with advanced pNET is critical, particularly in the situation of donor organ shortage.

#### 3.3.3. pNET with Extensive Liver Metastasis (>50%) or Extrahepatic Disease

In this setting, most NET dedicated tumor centers would initialize treatment with SSA’s or PRRT in patients with a high demand for aggressive therapy. Patients with a response to therapy or a stable course over time might qualify for a more surgical approach. However, recurrence is likely, and calculation of post-surgery steps should be taken into consideration, including either surveillance or continuation of systemic treatment. This is also true for locoregional strategies, e.g., SIRT, where limited data is available that this is feasible in highly selected cases [99].

An important question in patients with a high disease burden is how to deal with the primary tumor, which is ambiguously discussed in the literature if the tumor is asymptomatic [18,100,101]. Although these retrospective studies should be interpreted with caution due to their potential bias, patients with low tumor burden and a good functional status may benefit from resection of the primary [102]. However, this question should be addressed by further studies. Uncontrolled extrahepatic disease, however, is an independent negative prognostic factor, and these patients should not undergo resection of an asymptomatic primary.

In patients with extensive liver metastasis, debulking or cytoreductive surgery has been proposed by several groups. In 2003, colleagues at the Mayo Clinic were one of the first groups to present their experience with liver resection and cytoreductive surgery in NET patients with NELM [84]. Half of these patients received major hepatic resection and symptom control was achieved in 96% of patients with initial NELM symptoms. Unfortunately, within 5 years, disease recurrence rate was reported by 84% of patients. Despite this, survival rates were promising, with 61% at 5 years and 35% at 10 years. The authors, therefore, concluded that at least 80% tumor debulking is necessary to demonstrate any survival outcome. Similar studies were presented with comparable results [12,39]. Negative predictive factors in these studies were patients with synchronous disease (hazard ratio 1.9), nonfunctional NET hormonal status (hazard ratio 2.0), and extrahepatic disease (hazard ratio 3.0) [39].

Although extrahepatic disease is associated with a worse prognosis in several series, patients with limited, stable extrahepatic disease can be considered for cytoreductive surgery, especially if NELM are symptomatic and debulking surgery would provide palliation of symptoms due to hormonal excess. In contrast, the role of cytoreductive surgery in non-secretory NET is controversial [83,84,103]. Data with a promising effect are retrospective and should be warranted with caution. In pNET, lung metastasis is relatively rare (around 5%) and usually goes together with progressive disease in the abdomen [104]. Resection of extrahepatic metastasis in low-grade pNET is technically possible and associated with acceptable outcomes in selected cases [17].

The addition of other modalities like SIRT for symptom relief is still debated in the cytoreductive setting. Although SIRT could achieve symptom control in NELM [105], a sequential approach following surgery should be avoided. In a study including 12 patients, liver surgery after SIRT was associated with increased morbidity and hospital readmission [19].

### 3.4. Timing Treatment Modalities in the Context of High-Grade Metastatic pNET—Allegro Ma Non-Troppo (Cheerful but Not Too Much)

Due to the current update of the pathologic classification by the WHO, only limited data exist for the handling of a G3 neuroendocrine neoplasm. The specific biologic behavior of a NEN G3 with a typical Ki-67 range from 20–55% differs highly from a NEN G2 or a neuroendocrine carcinoma (NEC) G3 [106]. Compared to the latter, G3 NENs have a better prognosis [106]. In patients with a resectable G3 NEN, surgery is possible after discussion in MDT, which should evaluate potential sequential steps that are well presented in the latest ESMO guidelines [14]. Radical surgery in G3 NEN with a Ki-67 < 55% showed a benefit in pancreatic NEN [107]. Retrospective data underline the benefit for surgery by 20% within the 3-year survival (69 vs. 49%) in a metastatic setting compared to systemic therapy only [108]. A predicting factor in this scenario might be the duration of control by the current systemic treatment. Currently, no prospective data exist for the prediction of the duration of control by systemic treatment. Herein, reaching durable systemic control by the initial treatment directly reaching in prolongation of OS up to 59 months in this dismal situation is possible [107]. Despite the lack of prospective controlled studies for this new entity, some retrospective data showed disease control under systemic treatment with the combination of capecitabine and temozolomide [109,110], as well as with everolimus [62] or streptozocin-based chemotherapy [111].

One scenario of a NEN G3 with surgical consideration might present by starting systemic therapy first and discussing tumor debulking or even the primary resection for symptom control. This scenario characterizes the personalized medicine approach, whereas the decision is based on the patient and tumor characteristics. Interestingly, this aspect is depicted in the latest ESMO guidelines of neuroendocrine gastro–enteropancreatic tumors [14]. Recently and within the upcoming years, data of new biomarkers, e.g., the impact of PD1/PD-L1, as well as the effectiveness of immune-oncology agents, are awaited as it seems that some subgroups of this heterogeneous p-NET G3 might be susceptible to immune therapy [112].

The DUNE trial confirmed the efficacy with durable responses of dual checkpoint blockade by durvalumab plus tremelimumab in the pancreatic G3 NEN population [113]. This will also open the discussion for neoadjuvant checkpoint blockade, where this highly effective concept is already confirmed in several solid tumors [114,115].

A final aspect in this challenging setting will be the discussion if single-agent respective combination strategies (e.g., PRRT and TKI or TKI with checkpoint blockade) should be given together or in a sequential approach. Luckily, recent recruitment studies will provide some insights to clinicians in the near future (e.g., NICE-NEC Study, NCT03980925; CABATEN Study, NCT04400474; AveNEC Study, NCT03352934). A special focus will be on improving disease control of NEN/NEC G3. The right timing of surgery in the scenario of NEN/NEC G3 is still unclear and will rely on better prognostic factors.

## 4. Conclusions

Surgery remains the mainstay in the sequential treatment of advanced pancreatic neuroendocrine tumors. Local therapies, such as TAE, TACE, or SIRT, might be evaluated in selected cases where the extrahepatic situation is controlled but systemic treatment has failed to control NELM or surgery is contraindicated. Timing of treatment modalities is highly affected by predictive and prognostic factors like the tumor burden [4] or the proliferation index Ki-67, where G2 and G3 NEN with a Ki67 < 55% should be considered for resection [107].

Upfront liver resection is preferred in low-volume metastasis or at least resectable disease, while liver transplantation is limited to patients with a favorable grading limited disease volume and who fulfill stringent selection criteria. Available systemic treatments, including PRRT, may be preferred as an alternative to upfront surgery to achieve a downsizing of the tumor and better disease control in borderline situations, to identify patients with benign tumor biology. Overall, there is limited data available on the precise timing of treatment modalities, and we highly recommend discussing treatment strategies at a dedicated MDT, preferentially at a NEN specialist center.

## Figures and Tables

**Table 1 cancers-14-01478-t001:** Grading of gastrointestinal neuroendocrine tumors by WHO 2017 classification [29].

	KI-67 Index (%)	Mitotic Index
Well-differentiated NENs		
NET G1	<3	<2/10 HPF
NET G2	3–20	2–20/10 HPF
Poorly differentiated NENs		
NEC G3	>20	>20/10 HPF
Small cell type		
Large cell type		
MINEN/MENEN		

Source: Adapted from WHO Classification of Tumors of Endocrine Organs, fourth edition (2017). Abbreviations: HPF, high-power field; MINEN/MENEN, mixed endocrine non-endocrine neoplasms; NEC, neuroendocrine carcinoma; NEN, neuroendocrine neoplasm; NET, neuroendocrine tumor; WHO, World Health Organization.

**Table 2 cancers-14-01478-t002:** Prognostic factors for surgery of advanced pancreatic NET in the metastatic setting.

Favorable Prognosis for Surgery	Unfavorable Prognosis for Surgery
Grading (WHO 2017): well differentiated low Grade G1 (Ki-67 < 3%) and Moderate Grade G2 (Ki-67 3–20%)	Grading (WHO 2017): well differentiated High grade NET G3 (Ki-67 > 20%) Poorly differentiated High grade NEC G3 (Ki-67 > 20%)
**T-Stage:** Any stage is favorable	
**N-Stage:** Locoregional N Stage within the surgical field of primary removal	**N-Stage:** Distant nodal involvement e.g., perihiliar nodal involvement, thoracic nodal involvement, infra- or para-aortic nodal involvement
**M-Stage:** Low volume and low count on metastasis **and** controlled by systemic treatment +/− sequential strategy of metastatic surgery	**M-Stage:** Disseminated metastatic situation in one or several organs +/− not controlled by systemic therapy
Performance status (ECOG PS 0-1)	Performance status (ECOG PS > 2)
**Factors without prognostic value:** age, gender, localization of pancreatic tumor (head, body, tail), lines of pre-treatment	

Abbreviations: ECOG PS, Eastern Cooperative Oncology Group Performance Status; NEC, neuroendocrine carcinoma; NEN, neuroendocrine neoplasm; NET, neuroendocrine tumor; WHO, World Health Organization.

**Table 3 cancers-14-01478-t003:** Systemic treatment with responses in advanced pNET.

	Intervention	n/n (Pancreas)	Grading	PFS (Months)	Survival 5 Years	Survival mOS (Months)	Pretreatment	Comments
CLARINET [65]	Lanreotide (Lan) vs. Placebo	204/91	G1-G2 (Ki67 < 10%)	NR vs. 18 ^#^	n/a	n/a	No systemic treatment, no major surgery allowed	Cross-over of placebo to Lanreotide was possible. At 2 y timepoint no significant between group differences in quality of life or overall survival were reported
RADIANT-3 [66]	Everolimus (Eve)	410	G1-G2	11 vs. 4.6	n/a	44 vs. 37.7	Antineoplastic treatment was allowed, but radiofrequency ablation or embolization of liver metastasis were excluded from study	Crossover from placebo to Eve allowed on disease progression
NETTER [67,68]	^177^LuDOTATATE vs. Placebo (continuous SSA)	229/none	G1-G2 (Ki67 < 20%)	28.35 vs. 8.74	n/a	48 vs. 36.3	Yes, at least with SSA	Cross-over allowed and 36% of placebo group patients received PRRT in cross-over
SUN-1111 [63,69]	Sunitinib vs. Placebo	171/160 completed trial	G1-G2	11.4 vs. 5.5	n/a	38.6 vs. 29.1	Yes, at least one prior treatment except prior TKI	SUN-1111 stopped early due to high rates of side effects. Cross-over from placebo to Sunitinib allowed
SANET-p [70]	Surufatinib vs. Placebo	172	G1-G2	10.9 vs. 3.7	n/a	Not yet reported	Yes, at least one but not more than two prior treatments (incl SSA, mTOR, PRRT)	Data from first interim analysis of 70% of reported PFS population
Strosberg et al., 2011 [64]	Capecitabine plus Temozolomid	30	G1-G2	18	n/a	92% at 2 years alive, 5-year survival not reported	Prior octreotide, interferon-α, or locoregional therapy with HAE were included	High ORR with 70%, only 4 patients (12%) with AE grade 3–4
TALENT [71]	Lenvatinib	111/55	G1-G2	15.6	n/a	32	Prior treatment with targeting agent in pNET group	Phase II study, median duration of response in pNET 19.9 months with disease control rate of 96.2%
Review PRRT in pNET [72]	^177^LuDOTATATE	Ranging from 29–68 pNET in a single study	G1-G2	Range 29–42	Not reported	Range 39 not reached	At least one prior line	Prospective and retrospective data analyzed in this review for efficacy of PRRT in pNET
Clewemar et al., 2015 [73]	STZ/5FU	133	G1-G3	23	Not reported	51.9	Yes and no	23.3% SSA 16.5% chemotherapy, 63.2% no prior treatment

# No subgroup analysis of pNET specific survival in these studies have been reported. Abbreviations: STZ, streptocozin; 5-FU, 5-fluorouracil; SSA, somatostatine analogue; N.R., not reached; pNET, pancreatic neuroendocrine tumor; PRRT, peptide-related therapy; ORR, objective response rate; HAE, hepatic artery embolization. The only exception, enabling a reasonable response rate, is peptide related radionucleatide therapy (PRRT), where results from a randomized study—the so-called NETTER-1 trial—demonstrated an 18% response rate according to RECIST criteria [67]. This study, however, included only midgut tumors excluding pNET. However, retrospective studies support the biological rationale to target a SSTR-2 positive pNET and provide data that PRRT is also effective in this setting [72,74]. For several reasons, it is crucial to consider the above-mentioned options ahead of surgery. First, STTR-2 targeting modalities with a downsizing effect like PRRT may induce a significant tumor response and may help to improve resectability. Second, minimal, non-visible disease may be treated by systemic modalities, reducing the risk of early recurrence, which is very common after resection of liver metastasis. Third, systemic treatment may allow for better assessment of the biology and behavior of the tumor, which may avoid unnecessary aggressive surgery and early recurrence.

**Table 4 cancers-14-01478-t004:** Outcomes for two-stage hepatectomy in patients with metastatic pNET.

	n/n (Pancreas)	Survival 5 Years	Survival mOS	Pretreatment	Comments
1995 Que [83]	74/unclear	73% at 4 years	N.R.	NR	No difference between curative resection and debulking
2010 Mayo [39]	339/134	74%	125 months	NR	Extrahepatic disease was poor prognostic factor
2003 Sarmiento [84]	170/52	61%			Complete resection in 75 (44%) patients
2018 Morgan [85]	42/42	81%	N.R.	NR	Proposed debulking threshold > 70%
2016 Maxwell [86]	108/28	76.1% (pNET)	10.5 years (pNET)	N.A.	Proposed debulking threshold > 70%
2019 Scott [87]	188/41	N.R.	N.R.	N.A.	>70% cytoreduction led to improved overall survival
2006 Musunuru [88]	48/15	83% (3 year)	N.R.	N.A.	Surgery is superior compared to non-surgical treatment

Abbreviations: N.A., not available; N.R., not reached.

**Table 5 cancers-14-01478-t005:** Overview of selected studies providing outcomes for liver transplantation in patients with metastatic pNET.

	n/n (Pancreas)	Recurrence	Survival 5 Years	Survival mOS	Pretreatment	Comments
2019, Korda 2019 [95]	10	50%	43%	N.A.	N.A.	all pNET (*n* = 3) recurred
2016, Mazzaferro [41]	42/15	13%	97%	N.R.	TACE/Resection	
2015, Sher [96]	85/42	56%	52%	N.A.	N.A.	20% multi-visceral TPL
2008, Le Treut [97]	85/(41)	N.A.	Around 25% in DP-NET	N.A.	N.A.	Hepatomegaly, pNET poor prognosis

Abbreviations: DP-NET, duodenal or pancreatic neuroendocrine tumor; N.R, not reached; N.A, not available; TPL, transplantation.

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
