# Peer review of "Orchestrating Treatment Modalities in Metastatic Pancreatic Neuroendocrine Tumors—Need for a Conductor"

_cancers, 2022, doi:10.3390/cancers14061478_

Round 1

Reviewer 1 Report

This is a comprehensive review of the management of metastatic pancreas neuroendocrine tumors.  There were a few grammatical and spelling errors which need revision.

I caution the authors on the language used when describing the role for liver transplantation in metastatic pancreas NET.  Contrastingly, there is inadequate discussion for the role of interventional radiology with locoregional liver therapies (e.g. TACE, y90, etc).

The musical references ("allegro") are appreciated by this reviewer, but I suspect many readers may find them distracting and potentially confusing.  

Finally, I believe a more formal meta-analysis would be more appropriate than this review.  

Author Response

Response to Reviewer 1 Comments

Point 1

This is a comprehensive review of the management of metastatic pancreas neuroendocrine tumors.  There were a few grammatical and spelling errors which need revision.

Response 1: We thank reviewer 1 for his evaluation and judgement of our review. We worked on the grammar and English within the revised version. However, we think to use the MDPI English editing service for a perfect presentation of our review if this is still needed.

Point 2

I caution the authors on the language used when describing the role for liver transplantation in metastatic pancreas NET.  Contrastingly, there is inadequate discussion for the role of interventional radiology with locoregional liver therapies (e.g. TACE, y90, etc).

Response 2: We thank reviewer 1 for his valuable statement in sight of liver transplantation. We have modified this passage. In addition we highlighted the role of interventional radiology and local therapies to liver metastasis.

Point 3

The musical references ("allegro") are appreciated by this reviewer, but I suspect many readers may find them distracting and potentially confusing.  

Response 3: We thank the reviewer 1 for adressing this point. We thought it would revitalize the paper if we combine the subheadings with some musicical references. However, we do not want to confuse the reader and have added the meaning in brackets now.

Point 4

Finally, I believe a more formal meta-analysis would be more appropriate than this review.  

Response 4: Bringing and analysing our points to a meta-analysis would be an approbiate step. However, as this was not a systematic review, no formal inclusion/exclusion criteria were selected. We cited studies that provided information regarding evaluation of surgery under systemic treatment as PRRT or MKIs or ICIs, focusing on downstaging and enabling surgery in this sequence in pNET. Thus, with focus on evaluation for surgery in different aspects in the sequence of pNET we thought that several studies would have been excluded from this narrative review although with their impact in a special focus (e.g. downstaging the tumor by PRRT) they are at that stage an important source to consider. In the end we did not change our setting to a meta-analysis and we are concinced that the current statement of our paper is still of high interest and valuable to evaluate surgery in pNET within different settings in advanced stages.

Reviewer 2 Report

Pancreatic neuroendocrine tumors are not recognized before the advanced stages when the tumor is hormonally active or the patient has symptoms related to extensive metastasis. In the review paper, the authors highlighted the time point for surgery for pancreatic neuroendocrine tumors. The topic is timely. The manuscript is well written. The text is clear and easy to read. I would still suggest a minor grammar and English language check before publication.

Author Response

Point 1

Pancreatic neuroendocrine tumors are not recognized before the advanced stages when the tumor is hormonally active or the patient has symptoms related to extensive metastasis. In the review paper, the authors highlighted the time point for surgery for pancreatic neuroendocrine tumors. The topic is timely. The manuscript is well written. The text is clear and easy to read. I would still suggest a minor grammar and English language check before publication.

Response 1: We thank reviewer 2 for his evaluation and judgement of our review. We worked on the grammar and English within the revised version. However, we think to use the MDPI English editing service for a perfect presentation of our review if this is still needed.

Round 2

Reviewer 1 Report

Appreciate the revisions.